# Solving Poisson Equations using Neural Walk-on-Spheres

**Hong Chul Nam**[*]
ETH Zurich
honam@student.ethz.ch

**Julius Berner**[*] **& Anima Anandkumar**
California Insitute of Technology
{jberner, anima}@caltech.edu

## Abstract

We propose *Neural Walk-on-Spheres* (NWoS), a novel neural PDE solver for the efficient solution of high-dimensional Poisson equations. Leveraging stochastic representations and Walk-on-Spheres methods, we develop novel losses for neural networks based on the recursive solution of Poisson equations on spheres inside the domain. The resulting method is highly parallelizable and does not require spatial gradients for the loss. We provide a comprehensive comparison against competing methods based on PINNs, the Deep Ritz method, and (backward) stochastic differential equations. In several challenging, high-dimensional numerical examples, we demonstrate the superiority of NWoS in terms of accuracy, speed, and computational costs. Compared to commonly used PINNs, our approach can reduce memory usage and errors by orders of magnitude. Furthermore, we apply NWoS to problems in the context of PDE-constrained optimization as well as molecular dynamics to show its efficiency in practical applications.

## 1 Introduction

Partial Differential Equations (PDE) are foundational to our modern scientific understanding in a wide range of domains. While decades of research have been devoted to this topic, numerical methods to solve PDEs still remain expensive for many PDEs. In recent years, deep learning has helped to accelerate the solution of PDEs (Azzizadenesheli et al., 2023; Zhang et al., 2023b; Cuomo et al., 2022) as well as tackle PDEs, which had been completely out of range for classical methods (Han et al., 2018; Scherbela et al., 2022; Nüsken & Richter, 2021b).

In this work, we focus on high-dimensional Poisson equations on general domains. Several deep learning methods (Raissi et al., 2019; Sirignano & Spiliopoulos, 2018; E et al., 2017; Han et al., 2017; Nüsken & Richter, 2021a; Han et al., 2020) are amendable to the numerical solution of Poisson equations. However, previous methods suffer from unnecessary high computational costs, bias, or instabilities, see Appendix A for a detailed comparison. To overcome these challenges, we propose a novel approach based on so-called *Walk-on-Spheres* (WoS) methods (Muller, 1956). WoS rewrites the solution as an expectation over Brownian motions stopped at the boundary of the domain. Leveraging the isotropy of Brownian motion, WoS accelerates the random walk by iterative sampling from spheres around the current position until reaching the boundary, see Figure 2.

**Our approach:** We develop *Neural Walk-on-Spheres* (NWoS), a version of WoS that can be combined with neural networks to learn the solution to (parametric families of) Poisson equations on the *whole domain*. Our method amortizes the cost of WoS during training so that the solution, as well as gradients, can be evaluated in fractions of seconds afterward (and at arbitrary points in the domain). The resulting objective is more efficient and scalable than competing methods, without the need to balance penalty terms for the boundary condition or compute spatial derivatives. In particular, we demonstrate a significant reduction of GPU memory usage in comparison to PINNs and up to orders of magnitude better performance for a given time and compute budget, see Table 1 and Figure 1.

**Related works:** We provide an in-depth comparison of competing deep learning approaches for the solution of elliptic PDEs in Appendix A. These include physics-informed neural networks

---

[*]Equal contribution

Table 1: Comparison of neural PDE solver for Poisson equations. *#derivatives*, *#loss terms*, and *cost* denote the order of spatial derivatives, the number of terms required in the loss function, and the computational cost for one gradient step. *Propagation speed* describes how quickly boundary information can propagate to the interior of the domain, see Appendix A for details.

| Method | #Derivatives | #Loss terms | Cost | Propagation speed |
|---|---|---|---|---|
| PINN | 2 | 2 | medium | slow |
| Deep Ritz | 1 | 2 | low | slow |
| Feynman-Kac | 0 | 1 | high | fast |
| BSDE | 1 | 1 | high | fast |
| Diffusion loss | 1 | 2 | medium | medium |
| **NWoS (ours)**[1] | **0** | **1** | **low** | **fast** |

(PINNs) (Raissi et al., 2019; Sirignano & Spiliopoulos, 2018), the Deep Ritz method (Jin et al., 2017), and the diffusion loss (Nüsken & Richter, 2021a), see also Table 1. The diffusion loss can be viewed as an interpolation between PINNs and losses based on backward SDEs (BSDEs) (Han et al., 2017; E et al., 2017; Beck et al., 2019). Methods based on BSDEs and the Feynman-Kac formula (Beck et al., 2018; Berner et al., 2020a; Richter & Berner, 2022) have been investigated for the solution of parabolic PDEs, where the SDE is stopped at a given terminal time. Due to costly simulation time, they cannot be applied efficiently to elliptic problems.

## 2 Neural Walk-on-Spheres (NWoS) Method

In this section, we derive our method. We consider the Poisson equation with Dirichlet boundary condition on an open, connected, and sufficiently regular domain $\Omega \subset \mathbb{R}^d$, given by

$$\begin{cases} \Delta u = f, & \text{on} \quad \Omega, \\ u = g, & \text{on} \quad \partial\Omega. \end{cases} \tag{1}$$

It is well-known that the solution admits the stochastic representation

$$u(x) = \mathbb{E}\left[g(X_\tau^\xi) - F_\tau^\xi \big| \xi = x\right], \quad \text{with} \quad F_\tau^\xi := \int_0^\tau f(X_t^\xi)\,\mathrm{d}t, \tag{2}$$

where $\xi$ is a random variable distributed on $\Omega$, $\mathrm{d}X_t = \sqrt{2}\,\mathrm{d}B_t$ is a scaled Brownian motion with initial condition $X_0^\xi = \xi$, and $\tau := \tau(\Omega, \xi) = \inf\{t \in [0, \infty) : X_t^\xi \neq \Omega\}$ is the first exit time, see Appendix A for further details. Directly leveraging this representation results in long runtimes, since it requires to simulate the Brownian motion until reaching the boundary. To tackle this issue, we cast the solution of the PDE in (1) into nested subproblems of solving Poisson equations on subdomains.

Specifically, let $\Omega_0 \subset \Omega$ be an open sub-domain containing[2] $\xi_0 := \xi$ and let $\tau_0 := \tau(\Omega_0, \xi)$ be the corresponding stopping time. Analogously to the stochastic representation in (2), we obtain that

$$u(\xi) = \mathbb{E}\left[u(X_{\tau_0}^\xi) - F_{\tau_0}^\xi \big| \xi\right]. \tag{3}$$

Note that this is a recursive definition since the solution $u$ to the PDE in (1) appears again in the expectation. To resolve the recurrence, we define the random variable $\xi_1 \sim X_{\tau_0}^\xi$ and choose another open sub-domain $\Omega_1 \subset \Omega$ containing $\xi_1$. Considering the stopping time $\tau_1 := \tau(\Omega_1, \xi_1)$, we can calculate the value of $u$ appearing in the expectation

$$u(X_{\tau_0}^\xi) \sim u(\xi_1) = \mathbb{E}\left[u(X_{\tau_1}^{\xi_1}) - F_{\tau_1}^{\xi_1} \big| \xi_1\right].$$

We can now iterate this process for $k \in \mathbb{N}$ and combine the result with (3) to obtain

$$u(\xi) = \mathbb{E}\left[g(X_\tau^\xi) - \sum_{k \geq 0} F_{\tau_k}^{\xi_k} \bigg| \xi\right]. \tag{4}$$

---

[1]While not necessary for NWoS, we note that the gradient of the model and an additional boundary loss can still be used to improve performance, see Appendix D.

[2]Since $\xi$ is a random variable, the sub-domain $\Omega_0$ is random, and the statement is to be understood for each realization.

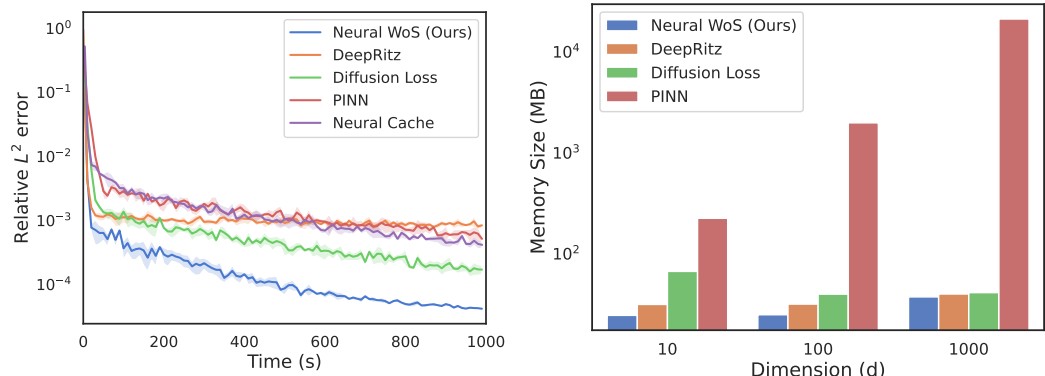

Figure 1: **Left:** Convergence of the relative $L^2$-error when solving the Laplace equation in $10d$ using our considered methods. **Right:** Peak GPU memory usage of different methods during training with batch size $512$ for the Poisson equation in Section 3 in different dimensions $d$.

In the above, we used the strong Markov property of the SDE solution as well as the tower property of the conditional expectation, see also Hermann et al. (2020).

## 2.1 WALK-ON-SPHERES

Picking $\Omega_k := B_{r_k}(\xi_k)$ to be a ball of radius $r_k \in (0, \infty)$ around $\xi_k$ in the $k$-th step, the isotropy of Brownian motion ensures that

$$\xi_{k+1} \sim X_{\tau_k}^{\xi_k} \sim \mathcal{U}(\partial B_{r_k}(\xi_k)).$$

In other words, we can just sample $\xi_{k+1}$ uniformly from a sphere of radius $r_k$ around the previous value $\xi_k$. To terminate after finitely many steps, we pick the maximal radius in each step, i.e., $r_k := \text{dist}(\xi_k, \partial\Omega)$, and stop when reaching an $\varepsilon$-shell, i.e., when $r_\kappa < \varepsilon$ for a prescribed $\varepsilon \in (0, \infty)$. This allows us to "walk" from sphere to sphere until (approximately) reaching the boundary, such that we can estimate the first term in (4). Specifically, the value $g(X_\tau^\xi)$ in (4) is approximated by the boundary value $g(\bar{\xi}_\kappa)$, where

$$\bar{\xi}_\kappa := \arg\min_{x \in \partial\Omega} \|x - \xi_\kappa\|$$

is the projection to the boundary, see Figure 2 and Figure 4 in the appendix. We note that the bias from introducing the stopping tolerance $\varepsilon$ can be estimated as $\mathcal{O}(\varepsilon)$ (Mascagni & Hwang, 2003). Moreover, for well-behaved, e.g., convex, domains $\Omega$, the average number of steps $\kappa$ behaves like $\mathcal{O}(\log(\varepsilon^{-1}))$ (Motoo, 1959; Binder & Braverman, 2012). This shows that $\varepsilon$ can be chosen sufficiently small without incurring too much additional computational cost. We note that this leads to much faster convergence than time-discretizations of the Brownian motion. In particular, to have a comparable bias, we would need to take steps of size $\mathcal{O}(\varepsilon)$, requiring $\Omega(\varepsilon^{-2})$ steps to converge.

Finally, we note that the second term in (4) can be estimated using Green's functions $G_{r_k}$ on the domains $\Omega_k = B_{r_k}(\xi_k)$, see Boggio (1905); Gazzola et al. (2010) and Appendix B.

## 2.2 LEARNING PROBLEM

Based on the previous derivations, we can establish a variational formulation, where the minimizer is guaranteed to approximate the Poisson equation in (1) on the whole domain $\Omega$. Specifically, we define

$$\mathcal{L}_{\text{NWoS}}[v] := \mathbb{E}\left[\left(v(\xi) - \text{WoS}(\xi)\right)^2\right], \tag{5}$$

where the single-trajectory WoS method $\text{WoS}(\xi)$ with random initial point $\xi$ is given by

$$\text{WoS}(\xi) := g(\bar{\xi}_\kappa) - \sum_{k=0}^{\kappa-1} |B_{r_k}(\xi_k)| f(\xi_k) G_{r_k}(\gamma_k, \xi_k).$$

In the above, $\gamma_k \sim \mathcal{U}(B_{r_k}(\xi_k))$, and the random variables $\kappa$, $\xi_k$, $\bar{\xi}_\kappa$, and $r_k$ are defined as in Section 2. From the stochastic formulation in (4) and Proposition 3.5 by Hermann et al. (2020), it follows that

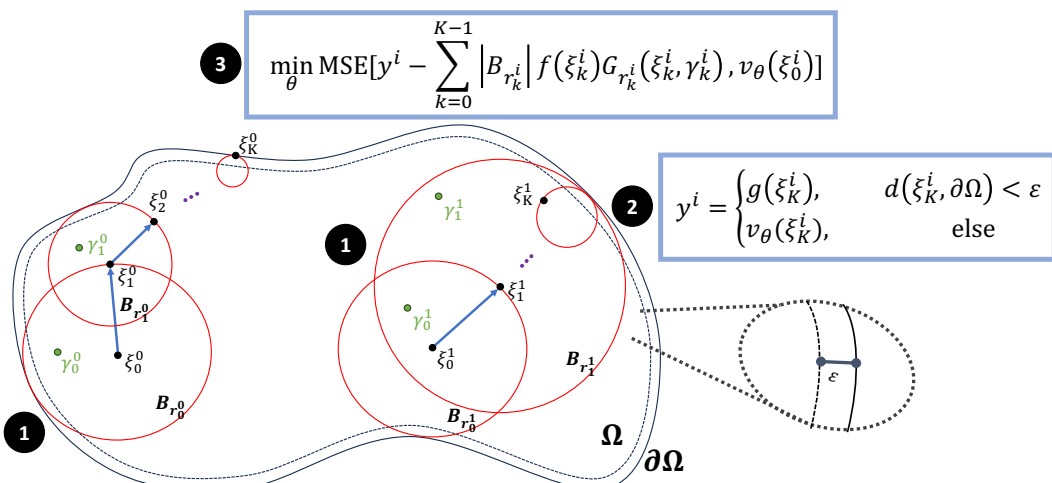

Figure 2: **Neural Walk-on-Spheres (NWoS):** Our algorithm for learning the solution to Poisson equations $\Delta u = f$ on $\Omega \subset \mathbb{R}^d$ and $u|_{\partial\Omega} = g$. **1** In each gradient descent step, we sample a batch of random points $(\xi_0^i)_{i=1}^m$ in the domain $\Omega$ and simulate Brownian motions by iteratively sampling $\xi_k^i$ from spheres $B_{r_k^i}$ inscribed in the domain. Moreover, we sample $\gamma_k^i \sim \mathcal{U}(B_{r_k^i})$ to compute a MC approximation $|B_{r_k^i}| f(\xi_k^i) G(\xi_k^i, \gamma_k^i)$ to the solution of the Poisson equation on the sphere $B_{r_k^i}$ using the Green's function $G$ in Appendix B. **2** We stop after a fixed number of maximum steps $\kappa$ and either evaluate our neural network $v_\theta$ or the boundary condition $g$ if we reach an $\varepsilon$-shell of $\partial\Omega$. **3** If $v_\theta$ satisfies the PDE, the mean-value property implies that $v_\theta(\xi_0^i)$ equals the expected value of $y^i$ minus the source term contributions. We thus minimize the corresponding mean squared error over the parameters $\theta$.

the minimizer of (5) approximates the solution $u$ in (1) in the uniform norm up to error $\mathcal{O}(\varepsilon)$, where $\varepsilon$ is the stopping tolerance. We refer to Appendix C for further theoretical results. We also remark that, in theory, the loss requires only a single WoS trajectory per sample of $\xi$ since the minimizer of the regression problem in (5) averages out the noise.

## 3 EXPERIMENTS

In this section, we compare the performance of NWoS, PINN, DeepRitz, diffusion loss, and Neural Cache (Li et al., 2023) on various baselines across dimensions from $10d$ to $50d$. We do not consider the FK and BSDE losses since they incur prohibitively long runtimes for simulating the SDEs with sufficient precision. To compare against the baselines, we consider benchmarks from the works proposing the Deep Ritz and diffusion loss Jin et al. (2017); Nüsken & Richter (2021a). To have a fair comparison, we use a fixed runtime of $25d + 750$ seconds and a GPU memory budget of 2GiB for training. Moreover, we ran a grid search over a series of hyperparameter configurations for each method. Then, we performed 5 independent runs for the best configurations w.r.t. the relative $L^2$-error. More details on the hyperparameters and our implementations[3] can be found in Appendix E.

We present our results in Table 2. We first note that we improve the Deep Ritz method as well as the diffusion loss by almost an order of magnitude compared to the results reported by Jin et al. (2017); Nüsken & Richter (2021a). Still, our NWoS approach can outperform all other methods on our considered benchmarks. In addition to these results, we highlight that the efficient objective of NWoS also leads to faster convergence, see Figure 1 and Appendix H. We provide ablation studies in Appendix G and additional numerical evidence for the Poisson equation in $100d$ and $500d$ in Appendix I.

We also highlight that the efficient objective of NWoS also leads to fast convergence and less memory footprint, see Figure 1. The PDE-constrained optimization problem shows that this allows NWoS to

---

[3]Our PyTorch code can be found at `https://github.com/bizoffermark/neural_wos`.

Table 2: Relative $L^2$-error (and standard deviations over 5 independent runs) of our considered methods, estimated using MC integration on $10^6$ uniformly distributed (unseen) points in $\Omega$.

| Method | Problem | | | |
| --- | --- | --- | --- | --- |
| | Laplace ($10d$) | Commitor ($10d$) | Poisson Rect. ($10d$) | Poisson ($50d$) |
| PINN | $7.42e^{-4} \pm 1.84e^{-4}$ | $4.10^{-3} \pm 1.11e^{-3}$ | $1.35e^{-2} \pm 1.57e^{-3}$ | $7.70e^{-3} \pm 2.25e^{-3}$ |
| Deep Ritz | $8.43e^{-4} \pm 6.29e^{-5}$ | $6.15e^{-3} \pm 5.30e^{-4}$ | $1.06e^{-2} \pm 6.20e^{-4}$ | $1.05e^{-3} \pm 1.70e^{-4}$ |
| Diffusion loss | $1.57e^{-4} \pm 7.74e^{-6}$ | $4.48e^{-2} \pm 6.93e^{-3}$ | $9.69e^{-2} \pm 1.03e^{-2}$ | $5.96e^{-4} \pm 1.06e^{-5}$ |
| Neural Cache | $3.99^{-4} \pm 4.08e^{-5}$ | $1.26e^{-3} \pm 5.82e^{-5}$ | $4.98e^{-2} \pm 1.80e^{-2}$ | $1.63e^{-2} \pm 1.42e^{-2}$ |
| WoS | $1.08e^{-3} \pm 1.34e^{-6}$ | $1.99e^{-3} \pm 9.79e^{-6}$ | $2.32e^{-1} \pm 2.09e^{-1}$ | $4.50e^{-3} \pm 7.38e^{-4}$ |
| **NWoS (ours)** | $\mathbf{4.29e^{-5} \pm 2.02e^{-6}}$ | $\mathbf{6.56e^{-4} \pm 2.42e^{-5}}$ | $\mathbf{2.60e^{-3} \pm 9.99e^{-5}}$ | $\mathbf{4.82e^{-4} \pm 1.32e^{-5}}$ |

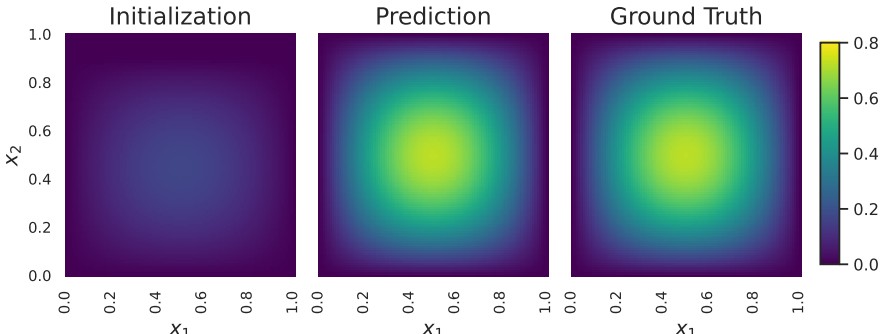

Figure 3: Qualitative assessment of the solution to the PDE-constrained optimization problem. **(Left)** Initial function $u_c$ for random parameters $c \in D$. **(Middle)** Predicted function $u_{\hat{c}}$ for the parameters $\hat{c}$ obtained after a few gradient descent steps using the approximation of the solution to the parametric Poisson equation obtained with NWoS. **(Middle)** The groundtruth solution $u_{c^*}$.

scale to parametric problems, where a whole family of Poisson equations is solved simultaneously. We observe that for this 5-dimensional problem (two spatial dimensions and three-parameter dimensions), NWoS converges within 20 minutes to a relative $L^2$-error of $0.79\%$ (averaged over $D \times \Omega$). More detailed results are shown in Appendix F.

In addition to solving PDE problems, we perform experiments on solving a trajectory optimization problem with PDE constraint, where the solution is to find the optimal control to achieve the target space while satisfying the PDE constraint. More detailed explanations are shown in Appendix F. It shows that NWoS can be extended to parametric problems, where a whole family of Poisson equations is solved simultaneously. We observe that for this 5-dimensional problem (two spatial dimensions and three-parameter dimensions), NWoS converges within 20 minutes to a relative $L^2$-error of $0.79\%$ (averaged over $D \times \Omega$). The trained network can then be used to solve the optimization problem directly (where we use L-BFGS) without requiring an inner loop for the PDE solver. The results show a promising relative $\ell^2$-error of $0.039\ \%$ for estimating the parameters $c^*$ leading to an accurate prediction of the minimizer, see Figure 3.

## 4 CONCLUSION

We have developed Neural Walk-on-Spheres, a novel way of solving high-dimensional Poisson equations using neural networks. Specifically, we provide a variational formulation with theoretical guarantees that amortizes the cost of the standard Walk-on-Spheres algorithm to learn solutions on the full underlying domain. The resulting estimator is more efficient than competing methods (PINNs, Deep Ritz method, Neural Cache, WoS, and diffusion loss) while achieving better performance at lower computational costs as well as faster convergence. We show that NWoS also performs better on a series of challenging, high-dimensional problems and parametric PDEs. This also highlights its potential for applications where such problems are prominent, e.g., in molecular dynamics and PDE-constraint optimization.

ACKNOWLEDGMENTS

J. Berner acknowledges support from the Wally Baer and Jeri Weiss Postdoctoral Fellowship. A. Anandkumar is supported in part by Bren endowed chair and by the AI2050 senior fellow program at Schmidt Sciences.

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

# A    NEURAL PDE SOLVER FOR ELLIPTIC PDES

We start by defining our problem as well as describing previous deep learning methods for its solution. Our goal is to approximate the solution[4] $u \in C(\Omega)$ to elliptic PDEs with Dirichlet boundary conditions of the form

$$\begin{cases} \mathcal{P}[u] = f, & \text{on} \quad \Omega, \\ u = g, & \text{on} \quad \partial\Omega, \end{cases} \tag{6}$$

with differential operator

$$\mathcal{P}[u] \coloneqq \tfrac{1}{2}\text{Tr}(\sigma\sigma^\top \text{Hess}_u) + \mu \cdot \nabla u.$$

In the above, $\Omega \subset \mathbb{R}^d$ is an open, bounded, connected, and sufficiently regular domain, see, e.g., Baldi (2017); Schilling & Partzsch (2014) for suitable regularity assumptions. Note that the formulation in (6) includes the Poisson equation 1 for $\mu = 0$ and $\sigma = \sqrt{2}\text{I}$.

In the following, we will summarize existing neural PDE solvers for these PDEs, see Table 1 for an overview. On a high level, they propose different variational formulations $\min_{v \in V} \mathcal{L}[v]$ with the property that every minimizer over a suitable function space $V \subset C(\Omega)$ is a solution $u$ to the PDE in (1). The space $V$ is then typically approximated by a set of neural networks with a given architecture, such that the minimization problem can be tackled using variants of stochastic gradient descent.

## A.1    STRONG AND WEAK FORMULATIONS OF ELLIPTIC PDES

Let us start with methods based on strong or weak formulations of the PDE in (6).

**Physics-informed neural networks (PINNs).**    In its basic form, the loss of PINNs (Raissi et al., 2019) or *Deep Galerkin* methods (Sirignano & Spiliopoulos, 2018), is given by

$$\mathcal{L}_{\text{PINN}}[v] \coloneqq \mathbb{E}\left[(\mathcal{P}[v](\xi) - f(\xi))^2\right] + \beta\mathcal{L}_{\text{bnd}}[v], \tag{7}$$

where

$$\mathcal{L}_{\text{bnd}}[v] \coloneqq \mathbb{E}\left[(v(\zeta) - g(\zeta))^2\right].$$

In the above, $\beta \in (0, \infty)$ is a penalty parameter, and $\xi$ and $\zeta$ are a suitable random variables distributed on $\Omega$ and $\partial\Omega$, respectively. While improved sampling methods have been investigated, see, e.g., Tang et al. (2023); Chen et al. (2023), the default choice is to pick uniform distributions. Given a set of samples, the expectations are approximated with standard MC estimators.

By minimizing the point-wise residual of the PDE, PINNs have gained popularity as a universal and simple method. However, PINNs are sensitive to hyperparameter choices, such as $\beta$, and suffer from training instabilities or high variance (Wang et al., 2021; Krishnapriyan et al., 2021; Nüsken & Richter, 2021b). Moreover, the objective in (7) requires the evaluation of the derivatives appearing in $\mathcal{P}[v]$. While this can be done exactly using automatic differentiation, it leads to high computational costs, see Figure 1.

**Deep Ritz method.**    For the Poisson equation in (1) one can avoid this cost by leveraging weak variational forms, see, e.g., Evans (2010). Rather than directly optimizing the regression loss in (7), the *Deep Ritz method* (E et al., 2017) proposes to minimize the objective

$$\mathcal{L}_{\text{Ritz}}[v] \coloneqq \mathbb{E}\left[\frac{\|\nabla v(\xi)\|^2}{2} - f(\xi)v(\xi)\right] + \beta\mathcal{L}_{\text{bnd}}[v]. \tag{8}$$

Under suitable assumptions, the minimizer again corresponds to the solution to the PDE in (1). However, the objective only requires computing the gradient $\nabla v$ instead of the Laplacian $\Delta v$. Using backward mode automatic differentiation, this reduces the number of backward passes from $d + 1$ to one, see also the reduced cost in Figure 1. Moreover, we note that the loss in (8) allows for weak solutions that are not twice differentiable. We refer to Chen et al. (2020), for an extensive comparison of the Deep Ritz method to PINNs for elliptic PDEs with different boundary conditions.

---

[4]For simplicity, we assume that a sufficiently smooth strong solution exists.

Finally, we mention that both methods suffer from the fact that the interior losses only consider local, pointwise information at $x \in \Omega$. At the beginning of the training, the interior loss might thus not be meaningful. Specifically, the boundary condition $g$ first needs to be learned via the boundary loss $\mathcal{L}_{\mathrm{bnd}}$, and then propagate from the boundary $\partial\Omega$ to an interior point $x$ via the local interior loss. There exist some heuristics to mitigate this issue by, e.g., progressively learning the solution, see Penwarden et al. (2023) for an overview. The next section describes more principled ways of including boundary information in the loss and directly informing the interior points of the boundary condition.

## A.2 Stochastic formulations of elliptic PDEs

From weak solutions, we will now proceed to stochastic representations of elliptic PDEs in (6). To this end, consider the[5] solution $X^\xi$ to the stochastic differential equation

$$\mathrm{d}X_t^\xi = \mu(X_t^\xi)\mathrm{d}t + \sigma(X_t^\xi)W_t, \quad X_0^\xi \sim \xi, \tag{9}$$

where $W$ is a standard $d$-dimensional Brownian motion. Moreover, we define the stopping time $\tau$ as the first exit time of the stochastic process $X^\xi$ from the domain $\Omega$, i.e.,

$$\tau = \tau(\Omega, \xi) := \inf\{t \in [0, \infty) \colon X_t^\xi \neq \Omega\}. \tag{10}$$

An application of Itô's lemma to the process $u(X_{t \wedge \tau}^\xi)$ shows that we almost surely have that

$$u(X_\tau^\xi) = u(X_0^\xi) + \int_0^\tau \mathcal{P}[u](X_t^\xi)\,\mathrm{d}t + S_\tau^u,$$

where $S_\tau^u$ is the stochastic integral

$$S_\tau^u := \int_0^\tau (\sigma^\top \nabla u)(X_t^\xi) \cdot \mathrm{d}W_t.$$

Using the fact that $X_0^\xi = \xi$ and assuming that $u$ solves the elliptic PDE in (6), we arrive at the formula

$$g(X_\tau^\xi) = u(\xi) + F_\tau^\xi + S_\tau^u, \tag{11}$$

where we used the abbreviation

$$F_\tau^\xi := \int_0^\tau f(X_t^\xi)\,\mathrm{d}t.$$

Since the stochastic integral $S_u^\tau$ has zero expectation, see, e.g., Baldi (2017, Theorem 10.2), we can rewrite (11) as a stochastic representation, i.e.,

$$u(x) = \mathbb{E}\left[g(X_\tau^\xi) - F_\tau^\xi \big| \xi = x\right], \tag{12}$$

which goes back to Kakutani's Theorem (Kakutani, 1944) and is a special case of the Feynman-Kac formula.

While the above representation allows us to establish MC methods for the pointwise approximation of $u$ at a given point $x \in \Omega$, it also allows us to derive variational formulation for learning $u$ on the whole domain $\Omega$. Based on the above results, we can derive the following three losses.

**Feynman-Kac loss.** The *Feynman-Kac* loss is given by

$$\mathcal{L}_{\mathrm{FK}}[v] := \mathbb{E}\left[\left(v(\xi) - g(X_\tau^\xi) + F_\tau^\xi\right)^2\right] \tag{13}$$

and follows from the fact that the solution to a quadratic regression problem as in (13) is given by the conditional expectation in (12). Notably, this variational formulation does neither require a derivative of the function $v$ nor an extra boundary loss $\mathcal{L}_{\mathrm{bnd}}$.

---

[5]We assume that there is a unique solution, see, e.g., Le Gall (2016) for corresponding conditions.

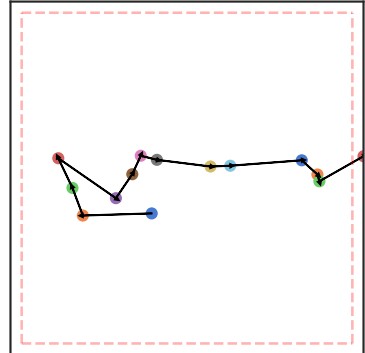 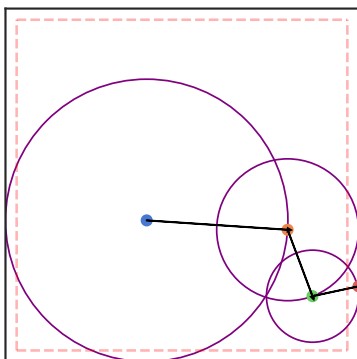

Figure 4: **Left:** Illustration of the solution $X^\xi$ to the SDE in (9) and its stopping time $\tau(\Omega, \xi)$ in (10) for the domain $\Omega = [0,1]^2$. **Right:** Realization of the Walk-on-Spheres algorithm in Section 2.

**BSDE loss.** Since the formula in (11) holds if and only if $u$ solves the PDE in (6), we can derive the *BSDE* loss

$$\mathcal{L}_{\mathrm{BSDE}}[v] := \mathbb{E}\left[\left(v(\xi) - g(X_\tau^\xi) + F_\tau^\xi + S_\tau^v\right)^2\right]. \tag{14}$$

Compared to the Feynman-Kac loss in (13), the BSDE loss requires computing the gradient of $v$ at every time discretization of the SDE $X^\xi$ in order to compute $S_v^\tau$. However, due to (11), $S_v^\tau$ acts as a control variate and causes the variance of the MC estimator of (14) to vanish at the optimum, see Richter & Berner (2022) for details.

For the previous two losses, boundary information is directly propagated along the trajectory of the SDE $X^\xi$ to the interior. However, simulating a batch of realizations of the SDE until they reach the boundary $\partial\Omega$, i.e., until the stopping time $\tau$, can incur prohibitively high costs.

**Diffusion loss.** The *diffusion loss* (Nüsken & Richter, 2021a) circumvents long simulation times by stopping the SDE at $s = \tau \wedge T$, i.e., at the minimum of a prescribed time $T \in (0, \infty)$ and the stopping time $\tau$. Since the trajectories might not reach the boundary, it is required to supplement the loss with a boundary loss. This yields the variational formulation

$$\mathcal{L}_{\mathrm{Diff}}[v] := \mathbb{E}\left[\left(v(\xi) - v(X_s^\xi) + F_s^\xi + S_s^v\right)^2\right] + \beta \mathcal{L}_{\mathrm{bnd}}[v].$$

Note that this can be viewed as an interpolation between the BSDE loss (for $s \to \infty$) and the PINN loss (for $s \to 0$ and rescaling by $s^{-2}$). In the same way, it also balances the advantages and disadvantages of both losses, see also Table 1.

## B    SOURCE TERM AND GREEN'S FUNCTION DERIVATION

To compute the second term in (4), we need to accumulate values of the form

$$v(z) := \mathbb{E}\left[-F_{\tau(B,z)}^z\right] \tag{15}$$

with a given ball $B = B_r(z)$. By (12), we observe that $v$ is just the solution of a Poisson equation on the ball $B$ with zero Dirichlet boundary condition evaluated at $z$.

By (12), we observe that $v$ is just the solution of a Poisson equation on the ball $B$ with zero Dirichlet boundary condition evaluated at $z$. We can thus use classical results by Boggio (1905), see also Gazzola et al. (2010), to write the solution in terms of Green's functions. Specifically, we have that

$$v(z) = -|B_r(z)| \, \mathbb{E}[f(\gamma) G_r(\gamma, z)], \tag{16}$$

where $\gamma \sim \mathcal{U}(B_r(z))$ and

$$G_r(y, z) := \begin{cases} \frac{1}{2\pi} \log \frac{r}{\|y-z\|}, & d = 2, \\ \frac{\Gamma(d/2-1)}{4\pi^{d/2}} \left(\|y - z\|^{2-d} - r^{2-d}\right), & d > 2, \end{cases}$$

In practice, we can now approximate the expectation in (16) using an MC estimate.

## B.1 DERIVATION

To compute integrals of the form (15), we look at a special case of a Poisson equation on a ball $B = B_r(z)$ with zero Dirichlet boundary condition, i.e.,

$$\begin{cases} \Delta v = f, & \text{on} \quad B, \\ v = 0, & \text{on} \quad \partial B. \end{cases}$$

Analogously to (12), we obtain that

$$v(z) = \mathbb{E}\left[ -\int_0^{\tau(B,z)} f(X_t^z)\,\mathrm{d}t \right], \tag{17}$$

where $\tau(B, z)$ is the corresponding stopping time, see (10). However, since we simplified the domain to a simple ball, we can write the solution in terms of Green's functions. Specifically, we have that

$$v(z) = -\int_B f(y)G_r(y, z)\,\mathrm{d}y \tag{18}$$

where

$$G_r(y, z) := \begin{cases} \frac{1}{2\pi}\log\frac{r}{\|y-z\|}, & d = 2, \\ \frac{\Gamma(d/2-1)}{4\pi^{d/2}}\left(\|y-z\|^{2-d} - r^{2-d}\right), & d > 2. \end{cases}$$

We note that (18) is equivalent to (16).

While this is a classical result by Boggio (1905), see also Gazzola et al. (2010), we will sketch a proof in the following. We consider the Laplace equation $\Delta\Phi_x = \delta_x$ for given $x \in \mathbb{R}^d$ in the distributional sense. It is well known that the fundamental solution $\Phi_x$ is given by

$$\Phi_x(y) = \begin{cases} \frac{1}{2\pi}\log\|y-x\|, & d = 2, \\ -\frac{\|y-x\|^{2-d}}{(d-2)\omega_d}, & d > 2, \end{cases}$$

where

$$\omega_d = |\partial B_1(0)| = \frac{2\pi^{d/2}}{\Gamma(d/2)} = \frac{4\pi^{d/2}}{(d-2)\Gamma(d/2-1)}$$

is the surface measure of the d-dimensional unit ball $B_1(0)$. Under suitable conditions, it further holds that the solution to (17) is given by

$$v(x) = \int_B f(y)\left(\Phi_x(y) - \phi_x(y)\right)\,\mathrm{d}y \tag{19}$$

for every $x \in B$, where the *corrector function* $\phi_x$ satisfies the Laplace equation

$$\begin{cases} \Delta\phi_x = 0, & \text{on} \quad B, \\ \phi_x = \Phi_x, & \text{on} \quad \partial B. \end{cases}$$

see Evans (2010, Chapter 2.2). Based on (12) and the fact that $\Phi_z$ is constant at the boundary of $B = z + B_r(0)$, we can compute the value of the corrector function $\phi_z$, i.e.,

$$\phi_z(y) = \mathbb{E}[\Phi_z(X_{\tau(B,y)}^y)] = \begin{cases} \frac{1}{2\pi}\log r, & d = 2, \\ -\frac{r^{2-d}}{(d-2)\omega_d}, & d > 2. \end{cases}$$

This shows that the value of the Green's function at the center $z$ of the ball $B$ is given by

$$\Phi_z(y) - \phi_z(y) = -G_r(y, z),$$

which, together with (19), establishes the claim.

## B.2 STABILIZING NUMERICAL ERRORS

For numerical stability, we directly compute the quantity $\tilde{G}_r(\gamma, z) := |B_r(z)|G_r(\gamma, z)$ in practice, as needed in (16). The volume of the hyper-sphere $|B_r(z)|$ is given by

$$|B_r(z)| = \frac{\pi^{d/2}}{\Gamma(d/2+1)}r^d,$$

such that we obtain

$$\tilde{G}_r(\gamma, z) := \begin{cases} \frac{r^2}{2}\log\frac{r}{\|\gamma-z\|}, & d = 2, \\ \frac{r^d}{d(d-2)}\left(\|\gamma-z\|^{2-d} - r^{2-d}\right), & d > 2. \end{cases}$$

## C  Theoretical Analysis of WoS

Having established the learning problem in Section 2.2, we can analyze both approximation and generalization errors. For the former, Hermann et al. (2020) bounded the size of neural networks $v_\theta$ to approximate the solution $u$ up to a given accuracy. In particular, the number of required parameters $\theta$ only scales polynomially in the dimension $d$ and the reciprocal accuracy, as long as the functions $f$, $g$, and $\mathrm{dist}(\cdot, \partial\Omega)$ can be efficiently approximated by neural networks.

One can then leverage results by Berner et al. (2020b) to show that also the generalization error does not underlie the curse of dimensionality when minimizing the empirical risk, i.e., a MC approximation of (5), over a suitable set of neural networks $v_\theta$. Specifically, the number of required samples of $\xi$ to guarantee that the empirical minimizer approximates the solution $u$ up to a given accuracy also scales only polynomially in the underlying dimension and the reciprocal accuracy.

## D  Improvements of NWoS

In this section, we discuss several strategies to trade off accuracy and computational cost, and to reduce the variance of MC estimators of $\mathcal{L}_{\mathrm{NWoS}}$ in 5.

**WoS with Maximum Number of Steps:**  For sufficiently regular geometries, the probability for a walk to take more than $k$ steps, is exponentially decaying in $k$ (Binder & Braverman, 2012). However, if a single walk in our batch needs significantly more steps, it slows down the overall training. We thus introduce a deterministic maximum number of steps $\kappa \in \mathbb{N}$. However, we do not want to introduce non-negligible bias by just projecting to the closest point on the boundary.

Instead, we want to enforce the mean-value property on subdomains of $\Omega$ based on our recursion in Section 2. We thus propose to use the model $v$ instead of the boundary condition $g$ if the walk does not converge after $\kappa$ steps, i.e., we define

$$y^{\xi,v} := \begin{cases} v(\xi_\kappa), & d(\xi_\kappa, \partial\Omega) > \varepsilon, \\ g(\bar{\xi}_\kappa), & \text{else.} \end{cases}$$

We can then replace the second term in (5) by

$$\mathrm{WoS}(\xi, v) := y^{\xi,v} - \sum_{k=0}^{\kappa-1} |B_{r_k}(\xi_k)| f(\xi_k) G_{r_k}(\gamma_k, \xi_k).$$

This helps to reduce the bias when $d(\xi_\kappa, \partial\Omega)$ is non-negligible while exploiting the faster convergence assuming that we obtain increasingly good approximations $v_\theta \approx u$ during training of a neural network $v_\theta$. Our approach bears similarity to the diffusion loss, see Appendix A, however, we do not need to use a time-discretization of the SDE.

**Boundary Loss:**  We find empirically that an additional boundary loss can further improve the performance of our method. While it is theoretically not required to converge to the correct solution, it can especially help for a smaller number $\kappa$ of maximum steps (see the previous paragraph). In general, we thus sample a fraction of the points on the boundary $\partial\Omega$ and optimize

$$\mathcal{L}_{\mathrm{NWoS}}[v] + \beta \mathcal{L}_{\mathrm{bnd}}[v],$$

where $\mathcal{L}_{\mathrm{bnd}}$ is defined[6] as in (7).

**Variance-Reduction:**  While not necessarily needed for the objective in (5), we can still compute multiple WoS trajectories $N \in \mathbb{N}$ per sample of $\xi$ to reduce the variance. This leads to the estimator

$$\widehat{\mathcal{L}}_{\mathrm{NWoS}}[v] := \frac{1}{m} \left( \sum_{i=1}^{m} v(\xi^i) - \frac{1}{N} \sum_{n=1}^{N} \mathrm{WoS}^n(\xi^i) \right),$$

---

[6]Note that $\mathcal{L}_{\mathrm{bnd}}$ can be interpreted as a special case of $\mathcal{L}_{\mathrm{NWoS}}$ where the WoS method directly terminates since the initial points are sampled on the boundary.

where $\xi^i$ are i.i.d. samples of $\xi$ and $\mathrm{WoS}^n(\xi^i)$ are i.i.d. samples of $\mathrm{WoS}(\xi^i)$, i.e., $N$ trajectories with the same initial point $\xi^i$, see (5). Note that we vectorize the WoS simulations across both the initial points as well as the trajectories, making our NWoS method highly parallelizable and scalable to the large batch sizes.

We further introduce control variates to reduce the variance of estimating $\mathrm{WoS}(x)$, where we focus on a fixed $x \in \Omega$ for the ease of presentation. Control variates seek to reduce the variance by using the estimator

$$\mathbb{E}\left[\mathrm{WoS}(x)\right] \approx \mathbb{E}[\delta] + \frac{1}{N} \sum_{i=1}^{N} \mathrm{WoS}^n(x) - \delta^n,$$

where $\delta^n$ are i.i.d. samples of a random variable $\delta$ with known expectation. Motivated by Sawhney & Crane (2020), we use an approximation of the first-order term of a Taylor series of $u$ in the direction of the first WoS step. We assume that $\nabla v_\theta$ provides an increasingly accurate approximation of the gradient $\nabla u$ during training and propose to use

$$\delta^n := \nabla v_\theta(x) \cdot (\xi_1^n - x),$$

where $\xi_1^n$ is the first step of $\mathrm{WoS}^n(x)$. In particular, $\xi_1^n \sim \mathcal{U}(\partial B_{r_1}(x))$ and thus $\mathbb{E}[\delta] = 0$ holds for any function $v_\theta$. While we need to compute the gradient $\nabla v_\theta(x)$ for the control variate, we mention that this operation can be detached from the computational graph. In particular, we do not need to compute the derivative of $\nabla v_\theta(x)$ w.r.t. to the parameters $\theta$ as is necessary for PINNs, the Deep Ritz method, the diffusion loss, and the BSDE loss. In Appendix G, we empirically show that the overhead of using the control variate is insignificant.

**Buffer:** Motivated by Li et al. (2023), we can use a buffer to cache training points $(\xi^{(i)}, \mathrm{WoS}(\xi^{(i)}))$. Since we only update the buffer after a given number of training steps, this accelerates the training. Note that this is not possible for the other methods since they require evaluation of the current model. In every buffer update, we average over further trajectories for a fraction of points to improve their accuracy. However, different from Li et al. (2023), we also evict a fraction of points from the buffer and replace them with WoS estimates on newly sampled points in the domain $\Omega$ to balance the diversity and accuracy of the training data.

## E  IMPLEMENTATION DETAILS

We implement all methods in PyTorch and provide pseudocode in Algorithms 1 and 2. The experiments have been conducted on A100 GPUs.

For all our training, we use the Adam optimizer and limit the runtime to $25d + 750$ seconds for a fair comparison. In every step, we sample uniformly distributed samples $(\xi, \zeta)$ in the domain $\Omega$ and on the boundary $\partial\Omega$ to approximate the expectations of the loss and boundary terms. Moreover, we employ an exponentially decaying learning rate, which reduces the initial learning rate by two orders of magnitude throughout training. We choose a feedforward neural network with residual connections, 6 layers, and width 256, and GELU activation function. We also perform the grid search for the boundary loss penalty term, i.e.,

$$\beta \in \{0.5, 1, 5, 50, 100, 500, 1000, 5000\}.$$

We further include the batch size $m \in \{2^i\}_{i=7}^{17}$ in our grid-search. To have a fair comparison, we set a fixed GPU memory budget of 2GiB for training, leading to different maximal batches for network sizes and methods, see also Figure 1. Let us detail the hyperparameter choices specific to each method in the following.

**Neural Walk-on-Spheres (NWoS):** For all experiments with NWoS, we set $\varepsilon = 10^{-4}$. If using a boundary loss, we sweep over $\{0.1, 0.2, 0.3, 0.4, 0.5\}$ in the grid search to find the optimal proportion of the batch size for the boundary loss. We choose to do a grid search on control variate, neural target, and buffer to ensure that a combination with the best performance is chosen. We fix the buffer size to 10 times that of the batch size, and we sweep the frequency to update buffers over the grid of $\{10, 100, 1000\}$. We also sweep the max step in $\{0, 1, 5, 10, 50, 100\}$ and the number of trajectories in $\{1, 10, 100, 200, 300, 400, 500, 1000\}$.

---

**Algorithm 1** Training of our NWoS method

---

**Input:** neural network $v_\theta$ with initial parameters $\theta$, optimizer method step for updating the parameters, WoS method WoS in Algorithm 2, maximum number of iterations $T$, batch size $m, n, b$ for points in domain, boundary and buffer, buffer $\mathcal{B}$ of size $B$, boundary function $g$, buffer update period $L$, boundary term coefficient $\beta$

**Output:** optimized parameters $\theta$

    $\mathcal{B} \leftarrow \{(x_i, g(x_i))\}_{i=1}^B$                                  ▷ Sample points in $\partial\Omega$ and
                                                           ▷ initialize the buffer

    **for** $k \leftarrow 0, \ldots, T$ **do**
        **if** $k \mod L == 0$ **then**
            $x_\Omega \leftarrow$ sample from $\xi^{\otimes m}$                            ▷ Sample points in $\Omega$
            $x_\mathcal{B} \leftarrow$ sample from $\mathcal{B}$                             ▷ Samples points in $\mathcal{B}$
            $x \leftarrow [x_\Omega, x_\mathcal{B}]$                                  ▷ Concatenate points
            $[y_\Omega, y_\mathcal{B}] \leftarrow \text{vmap}[\text{WoS}(x, v_\theta)]$                         ▷ WoS
            $\mathcal{B} \leftarrow$ update with $(x_\mathcal{B}, y_\mathcal{B})$                    ▷ Update the buffer
            $\mathcal{B} \leftarrow$ replace with $(x_\Omega, y_\Omega)$                  ▷ Replace the buffer
        **end if**
        $x_{\partial\Omega} \leftarrow$ sample from $\zeta^{\otimes n}$                               ▷ Sample points in $\partial\Omega$
        $(x_\mathcal{B}, y_\mathcal{B}) \leftarrow$ sample from $\mathcal{B}$                        ▷ Sample points in $\mathcal{B}$
        $\widehat{\mathcal{L}}_\Omega = \text{MSE}(v_\theta(x_\mathcal{B}), y_B)$                              ▷ Domain loss
        $\mathcal{L}_{\partial\Omega} = \text{MSE}(v_\theta(x_{\partial\Omega}), g(x_{\partial\Omega}))$                  ▷ Boundary loss
        $\widehat{\mathcal{L}}_{\text{NWoS}} = \widehat{\mathcal{L}}_\Omega + \beta\widehat{\mathcal{L}}_{\partial\Omega}$                           ▷ Compute loss
        $\theta \leftarrow \text{step}\left(\gamma, \nabla_\theta\widehat{\mathcal{L}}_{\text{NWoS}}\right)$                           ▷ SGD step
    **end for**

---

**Algorithm 2** Walk-on-Spheres (WoS)

---

**Input:** neural network $v_\theta$ at step $k$, source term $f$, boundary term $g$, point for evaluation $x$, maximum number of steps $\kappa$, stopping tolerance $\varepsilon$, number of trajectories $N$

**Output:** estimator $\widehat{v}$ of solution $v$ to PDE in (1) at $x$

    $\hat{v} \leftarrow 0$
    **for** $i \leftarrow 1, \ldots, N$ **do**                                       ▷ Batched in implementation
        $s \leftarrow 0$
        **for** $k \leftarrow 0, \ldots, \kappa - 1$ **do**
            $r \leftarrow \text{dist}(x, \partial\Omega)$                              ▷ Compute distance to $\partial\Omega$
            **if** $r < \varepsilon$ **then**
                Break                                  ▷ Reach boundary
            **end if**
            $\gamma \leftarrow$ sample from $\mathcal{U}(B_r(x))$                      ▷ Estimate source
            $s \leftarrow s + |B_r(x)| f(x) G_r(x, \gamma)$
            $u \leftarrow$ sample from $\mathcal{U}(\partial B_r(x))$
            **if** $k = 0$ & use_control_variate **then**
                $s \leftarrow s - \nabla_x v_\theta(x) \cdot u$                   ▷ Variance reduction
            **end if**
            $x \leftarrow x + u$                                  ▷ Walk to next point
        **end for**
        **if** $\text{dist}(x, \partial\Omega) < \varepsilon$ **then**                    ▷ Estimate solution at $x$
            $\widehat{v} \leftarrow s + g(x)$
        **else**
            $\widehat{v} \leftarrow s + v_\theta(x)$
        **end if**
    **end for**
    $\hat{v} \leftarrow \frac{1}{N}\hat{v}$                                            ▷ Take the average

---

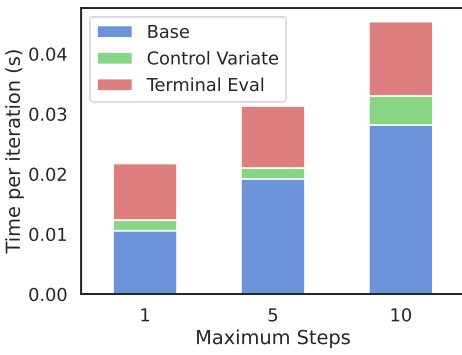 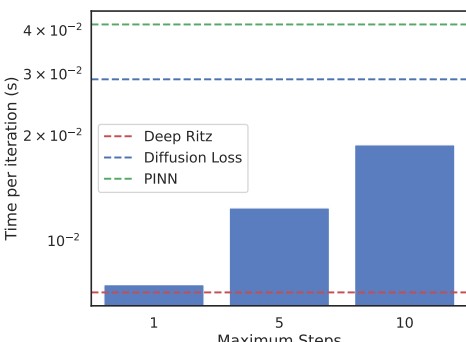

Figure 5: **Left:** Decomposition of the time for training one iteration of NWoS in the plain version (Section 2.2), as well as using our improvements from Appendix D, i.e., the control variate and a neural network evaluation for trajectories that did not converge in the given maximum number of steps $\kappa$. **Right:** Time for training one iteration for all considered methods. For NWoS we present the comparison for different maximum numbers of steps $\kappa$.

**Diffusion loss:**  For the diffusion loss (Nüsken & Richter, 2021a), also set $\varepsilon = 10^{-4}$. Moreover, we perform grid search for max step $s \in \{1, 5, 10, 50\}$ and time step $\Delta t \in \{10^{-3}, 10^{-4}, 10^{-5}\}$. We use 10% of the batch size for boundary points.

**PINNs:**  For PINNs (Raissi et al., 2019; Sirignano & Spiliopoulos, 2018), we use automatic differentiation to compute the Laplacian $\Delta v_\theta$. We use 10% of the batch size for boundary points.

**Deep Ritz:**  For the Deep Ritz method (E et al., 2017), we additionally experiment with the network architecture proposed in their paper. We sweep the number of blocks in $\{4, 6, 8\}$, the number of layers in $\{2, 4\}$, and the hidden dimension in $\{64, 128, 256\}$. Moreover, we observed better performance using the GELU activation function. We use 10% of the batch size for boundary points.

**Neural Cache:**  For the neural cache method (Li et al., 2023), we also search cache size over the grid of $\{10000, 20000, 100000, 1000000\}$, training period (i.e. period to update the cache) over $\{1, 10, 100, 1000, 5000, 10000\}$, the number of trajectories over $\{1, 10, 20, 30, 40, 50, 100, 500, 1000\}$.

**WoS:**  For Walk-on-Spheres (Muller, 1956), we directly approximate the solution at the evaluation points. We batch trajectories to saturate the memory budget and present the best result for different configurations within the given runtime. Specifically, we pick the number of trajectories in the grid $\{1, 10, 100, 1000, 10000, 100000\}$ and the maximum step in $\{0, 1, 10, 100, 1000\}$.

## F  EXPERIMENT DETAILS

Let us describe our considered PDEs in the following.

**Laplace Equation:**  The first PDE is a Laplace equation on a square domain given by

$$f(x) = 0, \quad g(x) = \sum_{i=0}^{d/2} x_{2i} x_{2i+1}, \quad x \in \Omega = (0,1)^d.$$

To test our models, we compare against the analytic solution as $u(x) = \sum_{i=0}^{d/2} x_{2k} x_{2k+1}$. Following Jin et al. (2017), we consider the case $d = 10$.

**Poisson Equation:**  Next, we consider the Poisson equation presented in Jin et al. (2017), i.e.,

$$f(x) = 2d, \quad g(x) = \sum_{i=1}^{d} x_i^2, \quad x \in \Omega = (0,1)^d,$$

with analytic solution $u(x) = \sum_{i=1}^{d} x_i^2$. We choose $d = 50$ and present results with[7] $d \in \{100, 500\}$ in Appendix I.

**Poisson Equation with Rectangular Annulus:** We also consider a Poisson equation on a rectangular annulus

$$\Omega = [-1, 1]^d \setminus [-c, c]^d$$

with sinusoidal boundary condition and source term

$$g(x) = \frac{1}{d} \sum_{i=1}^{d} \sin(2\pi x_i), \quad f(x) = -\frac{4\pi^2}{d} \sum_{i=1}^{d} \sin(2\pi x_i).$$

We choose $c = 0.25^{\frac{1}{d}}$ and $d = 10$ and note that the analytic solution is given by $u(x) = \frac{1}{d} \sum_{i=1}^{d} \sin(2\pi x_i)$.

**Committor Function:** The fourth equation deals with *committor functions* from molecular dynamics. These functions specify likely transition pathways as well as transition rates between (potentially metastable) regions or conformations of interest (Vanden-Eijnden et al., 2006; Lu & Nolen, 2015). They are typically high-dimensional and known to be challenging to compute. To compare NWoS, we consider the setting in Nüsken & Richter (2021a). The task is to estimate the probability of a particle hitting the outer surface of an annulus

$$\Omega = \{x \in \mathbb{R}^d : a < \|x\| < b\}, \quad a, b \in (0, \infty),$$

before the inner surface. The problem can then be formulated as the Laplace equation given by

$$f(x) = 0, \quad g(x) = 1_{\{\|x\|=b\}}, \quad x \in \Omega.$$

For this specific $\Omega$, a reference solution can be computed as

$$u(x) = \frac{a^2 - \|x\|^{2-d} a^2}{a^2 - b^{2-d} a^2}.$$

We further use the setting by Nüsken & Richter (2021a) and choose $a = 1$, $b = 2$, and $d = 10$.

**PDE-Constrained Optimization:** We want to solve the optimization problem

$$\min_{u \in H_0^1(\Omega), m \in L^2(\Omega)} \quad \frac{1}{2} \int_\Omega (u - u_d)^2 dx + \frac{\alpha}{2} \int_\Omega m^2 dx$$

constraint to $u$ being a solution to the Poisson equation with $g(x) = 0$ and $f(x) = -m(x)$ for $x \in \Omega = [0.1]^2$. The goal of the optimization problem is to balance the energy of the input control $m$ with the proximity of the state $u$ and the target state $u_d$, while satisfying the PDE constraint. Following Hwang et al. (2022), we choose $u_d = \frac{1}{2\pi} \sin(\pi x_1) \sin(\pi x_2)$ as target state.

To tackle this problem and showcase the capabilities of NWoS, we first solve a parametric Poisson problem, where we parametrize the control as $m_c = c_1 \sin(c_2 x_1) \sin(c_3 x_2)$ with $c \in D := [0.5, 1.0] \times [2.5, 3.5]^2$. Similar to Berner et al. (2020a), we can sample random $c \in D$ in every gradient descent step to use NWoS for solving a whole family of Poisson equations. Freezing the trained neural network parameters afterwards, we can reduce the PDE-constraint optimization problem to a problem over $c \in D$. In this illustrative example, we can compute the ground-truth parameters as $c^* = [\frac{1}{1+4\alpha\pi^4}, \pi, \pi]$ and choose $\alpha = 10^{-3}$.

# G ABLATION STUDIES

In this section, we analyze the speed of NWoS and perform comparisons with PINNs, Deep Ritz, and the diffusion loss. We assume the batch size to be fixed to $m = 512$ and test on the Poisson equation in Appendix F in 100d.

---

[7]While $d = 100$ is considered by Jin et al. (2017), we find that a simple projection outperforms all models in sufficiently high dimensions for this benchmark.

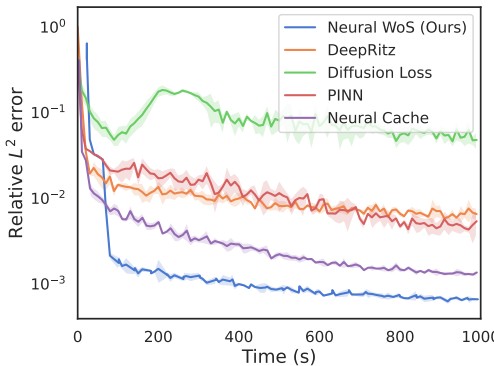

Figure 6: Convergence of the relative $L^2$-error when solving the Committor function in $10d$ using our considered methods.

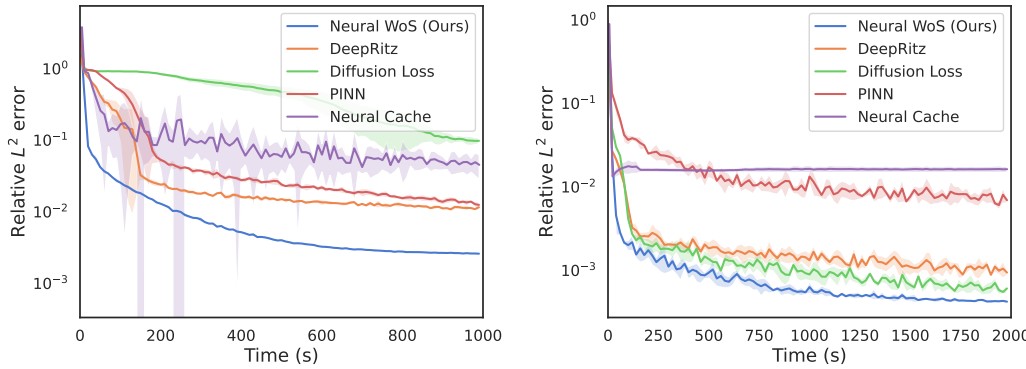

Figure 7: Convergence of the relative $L^2$-error when solving the Poisson equation in $10d$ on a rectangular torus (**left**) and the Poisson equation in $50d$ (**right**) using our considered methods.

Figure 5 decomposes the training time per iteration into the time for the base NWoS algorithm and the time for the additional features from Appendix D, namely the control variate as well as the neural network evaluation for a fixed number of steps $K$. Since both methods incur negligible overheads, we always used them in our experiments to reduce the bias and variance of our estimator.

Figure 5 shows the comparison of NWoS with DeepRitz, NSDE, and PINN with different maximum number of steps $K$, see Appendix E. Taking into account the logarithmic scaling of the plot, the training time of NWoS is significantly faster than both NSDE and PINN while slightly slower than Deep Ritz for higher maximum steps. In particular, the best results in Table Table 2 used $K = 10$ and outperformed Deep Ritz. However, choosing $K$ we can also trade-off between high-fidelity solutions and fast training.

## H    Convergence of the Relative $L^2$ Error

In this section, we illustrate the convergence of the relative $L^2$ error for other PDEs.

Figure 6 and 7 all demonstrate that neural WoS achieves the fastest convergence to the best optimum in comparison to all baseline methods provided time constraints.

## I    Further Evaluations

We further evaluate our method on the Poisson equation in $100d$ and $500d$ as proposed by E & Yu (2018). Table 3 demonstrates that our method is better than the baselines. However, we discover empirically that, for this benchmark, a simple projection to the boundary achieves the highest

Table 3: Relative $L^2$-error (and standard deviations over 5 independent runs) of our considered methods, estimated using MC integration on $10^6$ uniformly distributed (unseen) points in $\Omega$.

| Method | Problem | |
| --- | --- | --- |
| | Poisson (100$d$) | Poisson (500$d$) |
| PINN | $1.49e^{-3} \pm 3.21e^{-5}$ | $2.42^{-2} \pm 6.06e^{-4}$ |
| Deep Ritz | $1.77e^{-2} \pm 1.94e^{-4}$ | $9.92e^{-3} \pm 2.56e^{-5}$ |
| Diffusion loss | $6.71e-4 \pm 1.31e^{-5}$ | $9.47e^{-3} \pm 3.81e^{-5}$ |
| Projection | $\mathbf{2.92e^{-4} \pm 5.17e^{-7}}$ | $\mathbf{1.19e^{-5} \pm 1.67e^{-8}}$ |
| **NWoS (ours)** | $6.22e^{-4} \pm 1.18e^{-5}$ | $9.14^{-3} \pm 6.31e^{-5}$ |

accuracy. This can be motivated by the smoothness of the solution and the fact that uniformly distributed evaluation samples concentrate at the boundary in high dimensions.

## J  EXTENSIONS AND LIMITATIONS

NWoS is currently only applicable to Poisson equations with Dirichlet boundary conditions. While this PDE appears frequently in applications, we also believe that future work can extend our method. For instance, one can try to leverage WoS adaptations to spatially varying coefficients Sawhney et al. (2022), drift-diffusion problems Sabelfeld (2017), Neumann boundary conditions Sawhney et al. (2023); Simonov (2007), fractional Laplacians Kyprianou et al. (2018), the screened Poisson or Helmholtz equation Sawhney & Crane (2020); Cheshkova (1993), as well as linearized Poisson-Bolzmann equations Hwang & Mascagni (2001); Bossy et al. (2010). Moreover, one can also take other elementary shapes in each step, e.g., rectangles or stars Deaconu & Lejay (2006); Sawhney et al. (2023), and omit the need for $\varepsilon$-shells for certain geometries Given et al. (1997). using the Green's function first-passage algorithm Given et al. (1997).

Finally, while NWoS can tackle parametric PDEs, we need to have a fixed parametrization of the source or boundary functions. It would be promising to extend the ideas to neural operators, which currently only use losses based on PINNs Goswami et al. (2022); Li et al. (2021) or diffusion losses for parabolic PDEs Zhang et al. (2023a).

