# OpenReview forum: "Solving Poisson Equations using Neural Walk-on-Spheres"
_ICLR.cc/2024/Workshop/AI4DiffEqtnsInSci — AI4DiffEqtnsInSci @ ICLR 2024 Poster_

### Official Review · Reviewer_rtAb · 2024-02-24
**sound work**

**Rating:** 9
**Confidence:** 2

**Review:**

The proposed estimator effectively integrates boundary conditions without resorting to penalty terms, surpassing the efficiency of alternative methods like PINNs, the Deep Ritz method, and diffusion loss. The paper introduces innovative loss functions for neural networks, leveraging the Green function and recursive solution of Poisson equations within domain spheres, eliminating the need for computing spatial gradients in the loss calculation. The detailed implementation information provided in the appendix offers valuable guidance for readers. This paper demonstrates remarkable technical and mathematical rigor, presenting novel approaches and fully laying on the community's interest. I strongly recommend accepting this paper.

---

### Official Review · Reviewer_7wJr · 2024-02-25
**An interesting work, should be interesting for the community, minor revision.**

**Rating:** 7
**Confidence:** 2

**Review:**

The paper applies a novel strategy for solving high-dimensional Poison equations based on the neural walk-on-spheres technique. The approach and the results are interesting and significant, as I see. Also, I should admire the authors for staying loyal to the unanimous sharing of information, even their code. The paper looks interesting and novel to me. Also, the authors carefully explained the details of the work in most of the sections. They, then, compared the results to other approaches. I have below minor comments:
-	It would help clarify the approach if authors also mention the challenges of the method or cases in which it will have problems.
-	Is it possible to utilize neural WoS for inverse problems?
-	As far as I know, PINNs performance is really a function of choosing the right collocation points, and also a suitable training strategy. It would be better if the configurations that have been used for PINNs were discussed more carefully in appendix E (and maybe for other methods).

---

### Meta-Review · Area_Chair_GSR3 · 2024-03-01

**Recommendation:** Accept (Poster)

**Metareview:**

I appreciate reviewers' comments on this paper. Both agree that this paper proposes novel approach for solving high dimension Poisson Equation. Authors are required to address the comments and questions of reviewers in their final version. I vote accept for this paper.

---

### Decision · Program_Chairs · 2024-03-02

Accept (Poster)